# The Mother-Baby Bond: Role of Past and Current Relationships

**DOI:** 10.3390/children10030421

**Published:** 2023-02-22

**Authors:** Emanuela Bianciardi, Francesca Ongaretto, Alberto De Stefano, Alberto Siracusano, Cinzia Niolu

**Affiliations:** 1Department of Systems Medicine, University of Rome “Tor Vergata”, Via Cracovia, 50, 00133 Roma, Italy; 2Department of Surgical Sciences, Section of Gynecology and Obstetrics, University of Rome “Tor Vergata”, Via Montpellier, 1, 00133 Rome, Italy; 3Volunteers Association of Fondazione Policlinico “Tor Vergata”, 00133 Roma, Italy

**Keywords:** perinatal depression, attachment style, mother-infant relationship, mother-foetus bond, couple relationship, women health

## Abstract

During the perinatal period, up to 25% of women experience difficulties in relating to their child. The mother-child bond promotes the transition to motherhood, protects the woman from depression, and protects the child from the intergenerational transmission of the disease. This study prospectively investigated if the relationship with the co-parent, the attachment style, and the bond that women had with their parents influenced the mother-fetus and then mother-child bond. We also explored the role of depression and anxiety. One hundred nineteen pregnant women were enrolled. We administered clinical interviews and psychometric tools. A telephone interview was conducted at 1, 3, and 6 months of follow-up. Maternal insecure attachment style (r = −0.253, *p* = 0.006) and women’s dyadic adjustment in the couple’s relationships (r = 0.182, *p* = 0.049) were correlated with lower maternal–fetal attachment. Insecure attachment styles and depression correlate with bottle-feeding rather than breastfeeding. The bond women had with their mothers, not their fathers, was associated with breastfeeding. Depression (OR = 0.243, *p* = 0.008) and anxiety (OR = 0.185, *p* = 0.004; OR = 0.304, *p* < 0.0001) were related to mother-infant bonding. Close relationships, past and present, affect the bond with the fetus and the child differently. Psychotherapy can provide reassuring and restorative intersubjective experiences.

## 1. Introduction

It has been reported that in the perinatal period from 10 to 25% of women have relational problems with the unborn child characterized by feelings of anxiety and hostility towards the newborn up to the neglect and rejection of the child [1]. The mother-child relationship begins before birth and can facilitate the transition to motherhood, which is a critical physical, psychological and social transitional stage in women’s lives [2]. A strong mother-child bond is thought to protect women from depression and children from intergenerational transmission of the disease [3].

In line with the theory of prenatal attachment, “the fetus becomes more human to the woman as the pregnancy progresses, becoming loved both as an extension of itself and as an independent object” with which to form a relationship [4]. 

The mother-baby bond can be viewed as a bodily, immunological, perceptive, and affective relationship. The maternal physical change of pregnancy initiates the bodily relationship, which continues after delivery through mother’s care and breastfeeding.

The immunological relationship consisted of bi-directional communication determined on the one hand by fetal antigen presentation and on the other hand by recognition of and reaction to these antigens by the maternal immune system [5]. After delivery, it is required for the transmission of health and disease [6]. 

It is perceptible through the mother’s touch, fetal movements, ultrasound examinations, and eye-to-eye contact between mother and infant [7,8]. 

Finally, it has an impact on maternal emotional, brain, and behavioral changes that begin with conception [9,10]. 

A good mother-foetus relationship helps women take care of their pregnancy status by adopting healthy behaviors [11]. Moreover, the bond of the pregnant mother toward her unborn infant may be a predictor of an early mother-infant relationship postnatally [12]. 

According to the theory of the fetal origins of mental health [13], an individual’s mental and physical health status begin during intrauterine life [14]. Accordingly, maternal emotional well-being and the mother-foetus bond may influence neurodevelopmental outcomes in the offspring [15]. 

The maternal bond with the child laid the groundwork for secure attachment in offspring throughout the lifespan, promoted infant social-emotional development, and facilitated later parenting [16]. It is worth noting that the negative effects of prenatal adversity can be sensitive to the quality of the postnatal environment; in fact, the mother-infant relationship can be protective and reverse the course of brain dysfunction [17], as was clearly demonstrated in both animal and human studies [18]. Moreover, fathers can also protect the mother-child bond both by preventing maternal perinatal depression (PND) and by strengthening the couple relationship [19]. 

Furthermore, the risk of mental disorders in women’s lives increased during the difficult perinatal period, potentially having a negative impact on the mother-infant bond [20]. 

The aim of this prospective study was to deepen the understanding of the ongoing mother-baby relationship from pregnancy to the postpartum period. Prospectively, we explored how affective dimensions, such as anxiety and depression, and interpersonal functioning dimensions, such as attachment style (AS), couple relationships, and the bond that expectant mothers had with their parents influenced the relationship first with the fetus and then with the child. Furthermore, we investigated whether these personality and affective factors could account for the type and duration of breastfeeding.

## 2. Materials and Methods

The data for this study was collected as part of a larger longitudinal study of perinatal depression and infant development, which was advanced by the University of Rome “Tor Vergata” in Italy and promoted by the non-profit Volunteers Association of Tor Vergata Hospital organization. This data comes from an arm of the study that was conducted with the cooperation of the DSMDP ASL Roma 5 in Rome, Italy. From June 2018 to January 2019, 119 women were enrolled at the Mothers Clinics of the DSMDP ASL Roma 5 (Italy) Department of Gynecology and Obstetrics, which were affiliated with the University of Rome “Tor Vergata”. The inclusion criterion was being over 18 years old. The exclusion criteria were the diagnosis of psychotic disorders and insufficient knowledge of the Italian language. The study was conducted according to the standards of the Declaration of Helsinki and was approved by the Institutional Ethics Review Board of the University of Rome, Tor Vergata. All women signed informed consent. The study included four phases. In the first phase (T0), women were enrolled in the childbirth preparation course. A detailed, structured clinical interview and self-report questionnaires were administered. A telephone interview was conducted one month (T1), three months (T2), and six months (T3) after delivery to collect information about the delivery and the type of breast- or bottle-feeding (BF1st, BF3rd, BF6th). At the T1 session, participants completed the Mother-Infant Bonding Scale (MIBS) questionnaire via telephonic interview.

### 2.1. Structured Clinical Interview

The interview was administered by an obstetric nursing student (F.O.) with training relevant to perinatal depression. Questions were chosen based on the perinatal depression literature and included, but were not limited to, sociodemographic data and personal and family history of psychiatric disorders.

### 2.2. Edinburgh Postnatal Depression Scale

The Edinburgh Postnatal Depression Scale (EPDS) is a 10-item test that was originally developed to screen for postpartum depression but has since been adopted to screen for depression in pregnancy. Each question is scored from 0 to 3, and the total score ranges from 0 to 30, with higher values indicative of a more severe risk of depression. Scores of 14 or higher during pregnancy and 12 or higher after delivery have been shown to have the highest sensitivity and specificity for detecting depression. We used the validated Italian version of the EPDS [21].

### 2.3. Parental Bonding Instrument

The Parental Bonding Instrument (PBI) was used to measure parental behavior as perceived by the offspring. The instrument consists of 25 items: 12 “care” items and 13 “protection” items. Women are asked to rate their own parental behavior as they recall it from the first 16 years of their lives. There are 25 items each for the father figure and mother figure separately, with recalled child-rearing attitudes evaluated on a four-point (0–3) scale for 12 care items, 7 overprotection items, and 6 control items [22]. To date, there has been no consensus about the factor structure of the PBI. We extracted raw scores for each subscale with respect to the mother and father and analyzed them separately as follows: maternal care, maternal control, maternal overprotection, paternal care, paternal control, and paternal overprotection. Higher scores indicate preferred parenting attitudes in all dimensions.

### 2.4. Relationship Questionnaire 

Adult attachment style was assessed using the relationship questionnaire (RQ). Women were instructed to answer the questionnaire with reference to all their close relationships with peers (whether romantic or not). The RQ is a single-item measure comprising four short paragraphs, each describing a prototypical attachment pattern concerning close adult relationships. For each of the four descriptions, the respondents indicate how well it describes or relates to themselves on a seven-point rating scale. RQ provides a four-category model of AS based on the four combinations obtained by dichotomizing the subject’s mental representations of the self (self “internal working model” on one axis) and the subject’s image of the other (other “internal working model” on the orthogonal axis) into “positive” and “negative” based on their interpersonal relationships. This yields four attachment patterns: RQ1- secure (positive self, positive other), RQ2- preoccupied (negative self, positive other), RQ3- fearful (negative self, negative other), and RQ4- dismissing avoidant (positive self, negative other) [23,24]. 

### 2.5. State-Trait Anxiety Inventory (STAI) 

The state-trait anxiety inventory (STAI) investigates anxiety state and anxiety trait (forms Y1 and Y2, respectively) by means of 20 questions, scored on a scale from 1 to 4. A score of 0 to 29 indicates no anxiety, a score of 30 to 37 indicates mild anxiety, a score of 38 to 44 indicates moderate anxiety, and a score of > 44 indicates severe anxiety [25].

### 2.6. Dyadic Adjustment Scale (DAS) 

The Dyadic Adjustment Scale (DAS) is a 32-item, self-reported dyadic adjustment scale. This scale was designed to detect changes in the partner relationship and includes four scales: consensus (thirteen items), satisfaction (ten items), cohesion (five items), and affective expression (four items). The instrument also gives a total score of dyadic adjustment that ranges from 0 to 151. High total and subscale scores indicate a positive appraisal of the couple’s relationship. The internal consistency of the DAS total score is 0.96 [26]. 

### 2.7. Maternal Foetal Attachment Scale (MFAS)

The MFAS is a 24-item questionnaire organized into five subscales corresponding to aspects of the relationship between mother and fetus: differentiation of self from the fetus; interaction with the fetus; attributing characteristics and intentions to the fetus; giving of self; and role-taking. Women are rated on a five-point scale (from 1 absolutely no to 5 absolutely yes), and higher scores are associated with higher levels of maternal–fetal attachment [27].

### 2.8. Mother-to-Infant Bonding Scale (MIBS)

The Mother-to-Infant Bonding Scale (MIBS) is an 8-item self-report measure designed to assess a mother’s feelings towards her baby during the early postpartum period. Each item consists of an adjective (loving, resentful, protective, neutral or felt nothing, joyful, dislike, disappointed, aggressive) and is rated on a four-point scale [28].

### 2.9. Statistical Analyses 

We used descriptive analysis to study the frequency of dichotomous and continuous variables. Variables were treated either as continuous (MFAS total score, MIBS total score, RQ subscales, PBI subscales, age, STAI Y1-2 total score, EPDS total score, DAS total scores) or binary (couple relationship, employment, EPDS cut-off, RQ secure subscale versus RQ insecure subscales, breastfeeding at T1–BF1st, T2–BF3rd, and at T3–BF6th). 

We used the student’s t-test to compare the MFAS mean score in EPDS ≥ 12 versus EPDS < 12, and RQ = 1 versus RQ ≥ 1. The ANOVA test was used to compare the MFAS score among the four RQ1/RQ2/RQ3/RQ4 subscales of the RQ. A post hoc Games-Howell comparison and Kruskal-Wallis test were performed to compare the MFAS score to RQ1/2/3/4 subscales. 

We used the student’s t-test to analyze the differences between the groups of breastfeeding “no vs yes” at the T0, T1 and T3 times of the study (BF1st, BF3rd, BF6th) and the continuous variables (MFAS total score, Age, DAS total score, EPDS total score, STAI Y-1 total score, STAI Y-2 total score, RQ-1/2/3/4 subscales total score, PBI MOTHER CARE subscales total score, PBI MOTHER OVER PROTECTION subscales total score, PBI FATHER CARE subscales total score, PBI FATHER OVERPROTECTION subscales total score).

Bivariate correlations (Pearson r, two-tailed) were used to explore the association between the MFAS total score and the continuous variables that were listed above. We used an alpha level of 0.05 for significance (two-tailed). Correlation coefficients are considered to represent a small effect from 0.1 to 0.3, a medium effect from 0.3 to 0.5, and a large effect if greater than 0.5 [29]. 

Multiple linear regression, using the forward stepwise method, was used to test whether MFAS and MIBS could be predicted by the different variables. For dependent variables, standardized coefficients and regression coefficients beta were calculated. The statistical significance level was set a priori at *p* < 0.05 and calculations were done with the software IBM SPSS Statistics version 26 for Mac.

## 3. Results

The descriptive statistic is shown in Table 1 and Table 2. According to the EPDS score, 13.3% of women (16/119) suffered from depression during pregnancy. The prevalence of exclusive breastfeeding at the first month postpartum (BF1st), after three months (BF3th), and at the sixth month (BF6th) after delivery were 41.5, 40.1, and 28.4%, respectively.

The student’s *t*-test showed that the MFAS score was not different in women with and without depression according to the EPDS score. The ANOVA test revealed that the MFAS score was lower in the RQ3 group of women with a preoccupied attachment style (Table 3, Figure 1). 

As reported in Table 4, the correlation analyses (Pearson r two-tailed) demonstrated that the MFAS score was significantly and negatively related to the RQ- RQ3 fearful subscale (r = −0.253, *p* = 0.006). In addition, the DAS total score (r = 0.182, *p* = 0.049) was significantly and positively related to the MFAS score. 

Women who were not breastfeeding three months after partum had higher EPDS, lower PBI-mother care, and higher PBI-mother overprotection scores compared to those with exclusive breastfeeding (Table 5 and Table 6).

Women who were not breastfeeding six months after partum had higher EPDS scores, lower PBI-mother care scores, and were higher in the RQ3 subscale levels compared to those with exclusive breastfeeding (Table 7).

The first linear regression model (Table 8) demonstrated that RQ3 (*p* = 0.024, adjR^2^ = 0.037) was related to the MFAS score.

The second multiple linear regression model (Table 9) demonstrated that the EPDS (OR = 0.243, *p* = 0.008) and STAI Y1-2 (OR = 0.185, *p* = 0.004; OR = 0.304, *p* < 0.0001) total scores were related to the MIBS score.

## 4. Discussion

We found a high prevalence of depression in pregnant women, as has been extensively documented in the literature [30]. It should be noted that the women who participated in the study were not under psychotherapeutic or pharmacological treatment. Thus, we confirm that a large proportion of women with perinatal depression go undetected and untreated [31]. We discuss our findings in two blocks: those on the mother-fetus bond and those on the mother-child relationship.

The maternally insecure attachment style and women’s dyadic adjustment in the couple’s relationships were correlated with lower maternal–fetal attachment. 

Pregnancy represents a life event in which the couple dynamically adapts to change to achieve a new shared and lasting harmony. Failure to achieve this dyad adjustment is a risk factor for maternal-fetal bonding because the mother may perceive that the forthcoming baby is an obstacle to the couple’s relationship [32,33]. Moreover, difficulties in the couple’s relationship may constitute an additional risk factor, as it has been shown that the partner supports the mental health of the mother and the development of the child [19].

A person’s attachment style is determined by their own and others’ models of expectations, needs, feelings, and behaviors in close relationships. During pregnancy, the woman can face various stressors, including the pregnancy itself, which activates the attachment system [34]. While the adult attachment style is activated in close and bidirectional relationships, prenatal attachment is based on the unidirectional and abstract bond between mother and unborn child that is established during pregnancy [35]. For women with an insecure attachment style, mentally representing the unborn child, whose feelings and actions are less obvious than those of the flesh-and-blood child, can be especially difficult [36].

Depression and anxiety during pregnancy had no effect on the mother-fetus bond, contrary to what we expected [37]. However, the insecure attachment style has emerged as the best explanation for the weaker mother-fetus attachment, as well as the couple relationship [38]. 

In this study, we explored the mother-infant bond using two measures: a psychometric instrument that was completed by the women one month postpartum, and, in addition, we assessed the proportion of breastfed infants at one, three, and six months of age [39]. Our results demonstrated that depression in pregnancy was associated with bottle-feeding compared with breast-feeding at 1, 3, and 6 months postpartum, which can be explained either as a direct effect of depressive symptoms, such as anhedonia and fatigue, or an indirect effect of depression mediated by the poor mother-infant relationship [40]. In fact, we found that women with depression and anxiety during pregnancy rated their relationship with the baby worse.

Our findings confirm that perinatal depression has a negative impact on offspring, both through the depression itself and especially through the mother-infant relationship. This last aspect is fundamental because the newborn’s brain is not mature at the time of birth but develops rapidly through experiences [41] that the child has with the external environment, which is the mother, on whom it totally depends [42]. Interactions with the mother are therefore essential to modulate brain development and influence the child’s cognitive and emotional development. Furthermore, depression and anxiety in pregnancy affect fetal development through biological pathways, causing what has been defined as a “meta-plastic” brain state [13] in which sensitivity to external stimuli is heightened. The potential spectrum of consequences is dangerous when we consider that most mothers with depression go untreated. 

In our study, however, poor mother-fetal bonding was not associated with decreased mother-infant bonding or breastfeeding, as we would have expected [12].

We got some interesting results. First, we found that mothers with insecure attachment styles preferred bottle-feeding over breastfeeding at 1 month and 6 months postpartum (Table 5 and Table 7). The secure attachment style supports women’s interpersonal functioning, protects them from depression, and is associated with more effective emotional regulation strategies that improve the relationship with the child. Conversely, the insecure attachment style increased the risk of mood disorders, as bonding with the child can be perceived as a source of stress [43]. Furthermore, for the first time, we demonstrated that the relationship that the women in this study had with their parents in the first 15 years of their lives influenced their relationship with the newborn and breastfeeding. In particular, we found that the risk factor was poor maternal care and an overcontrolling attitude, characterized by intrusiveness and a limitation of autonomy. So, only the bond the women in the study had with their mothers, not their fathers, was significant. This figure highlights the primacy of the maternal line in women’s mental health. 

Moreover, it is important to consider that the literature studies have mainly been based on the attachment theory of John Bowlby [44], which refers to the emotional response of parents towards their children, paying less attention to the effects of overprotection and control by parents.

Furthermore, a continuity has been described between having had overcontrolling mothers and an insecure attachment style characterized by low self-esteem and a negative self-model [45]. From this perspective, the experience of a poor relationship with the mother may have determined negative internal models of self and others typical of the fearful attachment style, with the risk of depression and a negative impact on the bond with the child. 

Our results confirm that the perinatal phase can be conceptualized as a psychopathological continuum across generations. In fact, the intergenerational transmission—from the grandmother to the grandson who has not been directly exposed—and the transgenerational transmission—from the mother to the child—of insecure attachment could affect the children of the women in this study [16]. 

Finally, future research should look into the predictive value of an insecure attachment style not only in the mother-infant dyad but also in other attachment relationships such as the doctor-patient relationship and, as a result, treatment adherence [46,47].

Although the implications of our study are compelling, we recognize some limitations. To begin with, we could have explored attachment style using clinical interviews compared to self-report questionnaires. However, self-report tests are a reliable and widely accepted tool for research purposes [24]. Furthermore, we could have explored the couple’s dyadic adjustment from the partner’s point of view, considering that fathers protect the woman from depression and mitigate the negative effect of maternal depression on the offspring [48].

Besides these limitations, we highlighted the relevance of our study, which investigated the role of current and past relationships, attachment style, and maternal mental health on the antenatal and postnatal mother-baby bond. 

Maternal depression is amenable to treatment and thus is a modifiable risk factor to prevent poor child outcomes, although depression is not the only treatment target [49]. The International Guidelines on Women’s Mental Health recommend considering and promoting the mother-child bond [50,51]. 

As we have emphasized, dimensions of interpersonal functioning, such as the woman’s bonding with the mother, adjustment of the dyadic relationship with the partner, and attachment style, were essential for an adequate mother-infant relationship beyond perinatal depression. The psychotherapeutic path of women in the perinatal period should provide the conditions for accessing a reassuring and reparative intersubjective experience that responds to their needs. Furthermore, it would be appropriate to include the co-parents in the therapeutic project.

## Figures and Tables

**Figure 1 children-10-00421-f001:**
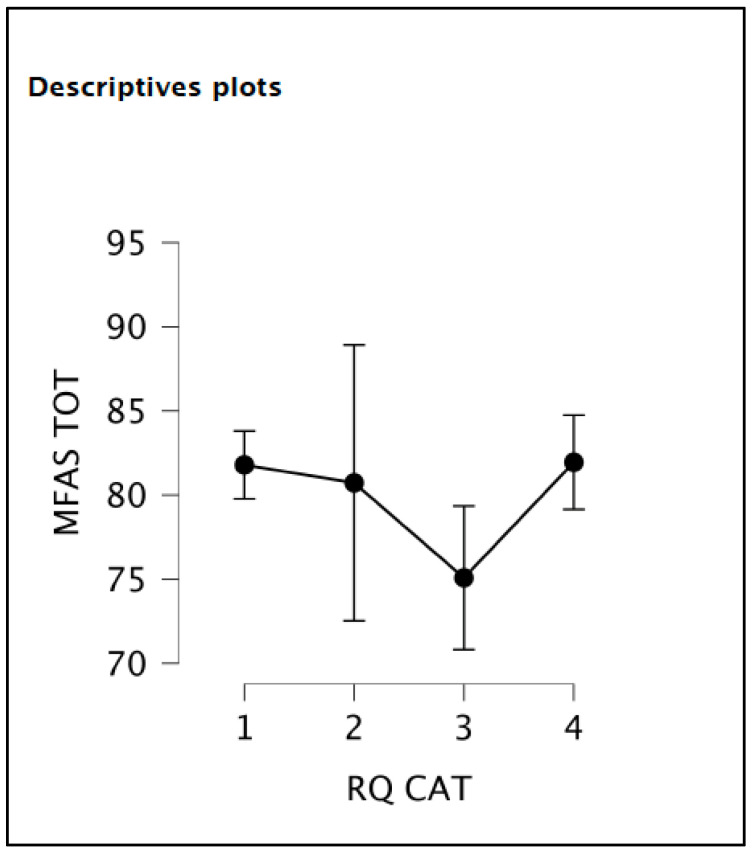
A descriptive plot of the MFAS mean score among the four RQ subscales.

**Table 1 children-10-00421-t001:** The demographics of the total sample (N = 119).

Variable	Value	N	%
Mean age yrs (range) SD		32.0 (18–46) SD = 5.30	
Education			
	Primary school	1	0.8
	Junior high school	22	18.5
	High school	58	48.8
	Bachelors degree/Post graduate	38	31.9
Couple relationship			
	Single/separated/divorced	3	2.5
	Married/cohabiting	116	97.5
Occupation			
	Unemployed	41	34.5
	Employed full/part time	78	65.5

SD: Standard deviation.

**Table 2 children-10-00421-t002:** The descriptive statistics of psychometric instruments.

Variable	Value	N	%
EPDS			
	<12	103	86.7
	≥12	16	13.3
RQ			
	Secure	65	54.6
	Preoccupied	7	5.9
	Fearful-Avoidant	14	11.8
	Dismissing-Avoidant	33	27.7
PBI Mother Care			
	>27	45	37.8
	<26	74	62.2
PBI Mother Protection			
	<13	64	53.7
	>13	55	46.2
PBI Father Care			
	<23	44	37.0
	>24	75	63.0
PBI Father Protection			
	<12	53	44.5
	>12.5	66	55.5
PBI Mother			
	Affectionate constraint	27	22.7
	Affectionate control	29	24.4
	Optimal parenting	48	40.3
	Neglectful parenting	15	12.6
PBI Father			
	Affectionate constraint	34	28.6
	Affectionate control	32	26.9
	Optimal parenting	41	34.4
	Neglectful parenting	12	10.1
Breastfeeding at 1st month (N = 118)	No 49		
	Yes 69	3	2.5
Breastfeeding at 3rd month (N = 117)	No 70		
	Yes 47	4	3.3
Breastfeeding at 6th month (N = 116)	No 83		
	Yes 33		

**Table 3 children-10-00421-t003:** ANOVA test, total MFAS versus categorial RQ.

Cases	Sum of Squares	df	Mean Square	F	*p*
RQ CAT Residuals	560.691 7277.173	3 114	186.897 63.835	2.928	0.037

Note. Type III Sum of Squares.

**Table 4 children-10-00421-t004:** Correlation analysis between MFAS score and age, DAS, EPDS, STAI Y 1-2, RQ1-4 subscales, and PBI subscales.

N			r	*p*
MFAS TOT	- AGE	118	−0.150	0.106
MFAS TOT	- DAS TOT	117	0.182 *	0.049
MFAS TOT	- EPDS (TOT)	118	0.069	0.458
MFAS TOT	- STAI Y-1 TOT	118	0.011	0.903
MFAS TOT	- RQ 1	118	0.044	0.633
MFAS TOT	- RQ 2	118	−0.088	0.345
MFAS TOT	- RQ 3	118	−0.253 **	0.006
MFAS TOT	- RQ 4	118	0.038	0.682
MFAS TOT	- PBI MOTHER CARE	118	0.053	0.572
MFAS TOT	- PBI MOTHER OVER PROTETION	118	−0.071	0.443
MFAS TOT	- PBI FATHER CARE	112	0.051	0.596
MFAS TOT	- PBI FATHER OVERPROTECTION	112	−0.128	0.180

* *p* < 0.05, ** *p* < 0.01.

**Table 5 children-10-00421-t005:** Independent samples *t*-test—continuous variables versus breastfeeding one month after partum (BF1st).

	Test	Statistic	df	*p*
MFAS TOT	Student	−0.108	114.000	0.914
	Welch	−0.108	109.709	0.914
	Mann-Whitney	1707.500		0.811
AGE	Student	−0.205	115.000	0.838
	Welch	−0.205	109.548	0.838
	Mann-Whitney	1662.500		0.882
DAS TOT	Student	−1.808	114.000	0.073
	Welch	−1.814	108.992	0.072
	Mann-Whitney	1320.500		0.061
EPDS (TOT)	Student	2.839	115.000	0.005
	Welch	2.955	112.624	0.004
	Mann-Whitney	2133.000		0.015
STAI Y-1 TOT	Student	1.442	115.000	0.152
	Welch	1.474	114.919	0.143
	Mann-Whitney	1887.000		0.281
STAI Y-2 TOT	Student	1.927	115.000	0.056
	Welch	1.960	114.361	0.052
	Mann-Whitney	1977.500		0.115
RQ 1	Student	−0.771	115.000	0.442
	Welch	−0.772	109.544	0.442
	Mann-Whitney	1522.500		0.352
RQ 2	Student	1.936	115.000	0.055
	Welch	1.965	113.967	0.052
	Mann-Whitney	2016.500		0.064
RQ 3	Student	0.508	115.000	0.613
	Welch	0.506	107.812	0.614
	Mann-Whitney	1809.000		0.504
RQ 4	Student	1.224	115.000	0.223
	Welch	1.213	105.170	0.228
	Mann-Whitney	1946.000		0.156
PBI MOTHER CARE	Student	−1.518	115.000	0.132
	Welch	−1.561	114.886	0.121
	Mann-Whitney	1473.000		0.234
PBI MOTHER OVER PROTECTION	Student	0.616	115.000	0.539
	Welch	0.624	113.839	0.534
	Mann-Whitney	1712.000		0.906
PBI FATHER CARE	Student	−1.317	109.000	0.191
	Welch	−1.296	96.013	0.198
	Mann-Whitney	1230.000		0.086
PBI FATHER OVERPROTECTION	Student	1.014	109.000	0.313
	Welch	1.037	108.759	0.302
	Mann-Whitney	1616.000		0.566

Bold text indicates significant variables.

**Table 6 children-10-00421-t006:** Independent samples t-test—continuous variables versus breastfeeding three months after partum (BF3rd).

	Test	Statistic	df	*p*
MFAS TOT	Student	−1.096	113.000	0.275
	Welch	−1.089	94.472	0.279
	Mann-Whitney	1430.500		0.373
AGE	Student	−0.218	114.000	0.828
	Welch	−0.210	85.271	0.834
	Mann-Whitney	1609.000		0.946
DAS TOT	Student	−1.572	113.000	0.119
	Welch	−1.593	103.537	0.114
	Mann-Whitney	1363.500		0.182
EPDS (TOT)	Student	3.977	114.000	<0.001
	Welch	3.627	68.009	<0.001
	Mann-Whitney	2199.000		0.001
STAI Y-1 TOT	Student	1.544	114.000	0.125
	Welch	1.434	73.570	0.156
	Mann-Whitney	1795.500		0.329
STAI Y-2 TOT	Student	1.085	114.000	0.280
	Welch	1.034	81.880	0.304
	Mann-Whitney	1733.500		0.530
RQ 1	Student	−0.880	114.000	0.381
	Welch	−0.848	85.612	0.399
	Mann-Whitney	1495.500		0.473
RQ 2	Student	1.374	114.000	0.172
	Welch	1.333	88.004	0.186
	Mann-Whitney	1825.000		0.237
RQ 3	Student	1.730	114.000	0.086
	Welch	1.724	97.709	0.088
	Mann-Whitney	1946.000		0.061
RQ 4	Student	1.552	114.000	0.124
	Welch	1.564	101.627	0.121
	Mann-Whitney	1910.500		0.100
PBI MOTHER CARE	Student	−2.460	114.000	0.015
	Welch	−2.341	81.376	0.022
	Mann-Whitney	1234.500		0.029
PBI MOTHER OVER PROTECTION	Student	2.076	114.000	0.040
	Welch	1.974	81.256	0.052
	Mann-Whitney	1907.500		0.108
PBI FATHER CARE	Student	−0.370	108.000	0.712
	Welch	−0.370	92.573	0.712
	Mann-Whitney	1402.500		0.764
PBI FATHER OVERPROTECTION	Student	1.326	108.000	0.188
	Welch	1.284	81.920	0.203
	Mann-Whitney	1636.000		0.262

**Table 7 children-10-00421-t007:** Independent samples t-test—continuous variables versus breastfeeding six months after partum (BF6th).

	Test	Statistic	df	*p*
MFAS TOT	Student	−1.375	113.000	0.172
	Welch	−1.373	109.545	0.173
	Mann-Whitney	1404.500		0.181
AGE	Student	0.104	114.000	0.917
	Welch	0.105	113.665	0.916
	Mann-Whitney	1720.500		0.779
DAS TOT	Student	−2.245	113.000	0.027
	Welch	−2.208	99.714	0.030
	Mann-Whitney	1237.000		0.024
EPDS (TOT)	Student	4.205	114.000	<0.001
	Welch	4.430	94.450	<0.001
	Mann-Whitney	2306.500		<0.001
STAI Y-1 TOT	Student	1.826	114.000	0.070
	Welch	1.881	110.655	0.063
	Mann-Whitney	1940.000		0.134
STAI Y-2 TOT	Student	1.346	114.000	0.181
	Welch	1.372	113.788	0.173
	Mann-Whitney	1885.000		0.233
RQ 1	Student	−0.828	114.000	0.409
	Welch	−0.839	113.899	0.403
	Mann-Whitney	1532.500		0.442
RQ 2	Student	1.326	114.000	0.187
	Welch	1.330	111.621	0.186
	Mann-Whitney	1933.000		0.131
RQ 3	Student	2.809	114.000	0.006
	Welch	2.845	113.912	0.005
	Mann-Whitney	2176.000		0.004
RQ 4	Student	0.743	114.000	0.459
	Welch	0.740	109.022	0.461
	Mann-Whitney	1817.500		0.408
PBI MOTHER CARE	Student	−2.506	114.000	0.014
	Welch	−2.569	112.460	0.012
	Mann-Whitney	1270.000		0.027
PBI MOTHER OVER PROTECTION	Student	1.627	114.000	0.107
	Welch	1.659	113.657	0.100
	Mann-Whitney	1907.500		0.187
PBI FATHER CARE	Student	−1.440	108.000	0.153
	Welch	−1.460	107.873	0.147
	Mann-Whitney	1288.000		0.203
PBI FATHER OVERPROTECTION	Student	0.926	108.000	0.357
	Welch	0.928	105.358	0.356
	Mann-Whitney	1645.500		0.383

**Table 8 children-10-00421-t008:** Linear regression (MFAS dependent variable).

Model Summary–MFAS TOT.									
**Model**	**R**	**R²**	**Adjusted R²**	**RMSE**	**R² Change**	**F Change**	**df1**	**df2**	** *p* **
1	0.000	0.000	0.000	8.139	0.000		0	110	
2	0.214	0.046	0.037	7.987	0.046	5.220	1	109	0.024
ANOVA									
**Model**			**Sum of Squares**		**df**	**Mean Square**	**F**		** *p* **
2	Regression		332.991		1	332.991	5.220		0.024
	Residual		6952.919		109	63.788			
	Total		7285.910		110				
Coefficients									
**Model**			**Unstandardized**		**Standard Error**	**Standardized**	**t**		** *p* **
1	(Intercept)		81.234		0.772		105.161		<0.001
2	(Intercept)		83.971		1.418		59.234		<0.001
	RQ 3		−0.999		0.437	−0.214	−2.285		0.024

Note for Model ANOVA. The intercept model is omitted, as no meaningful information can be shown. Note for Model Coefficients. The following covariates were considered but not included: AGE, DAS TOT, EPDS (TOT), STAI Y-1 TOT, STAI Y-2 TOT, AQ TOT, RQ 1, RQ 2, RQ 4, PBI MOTHER CARE, PBI MOTHER OVER PROTETION, PBI FATHER CARE, PBI FATHER OVERPROTECTION.

**Table 9 children-10-00421-t009:** Multiple linear regression (MIBS dependent variable).

Variable	Beta (a)	95%CI	SE	*p*
EPDS TOT	0.243	(0.003–0.022)	0.005	0.008
STAI Y-1	0.185	(0.002–0.009)	0.002	0.046
STAI Y-2	0.304	(0.004–0.014)	0.003	<0.0001

(a) Betas are standardized linear regression coefficients; correlation is significant at <0.05.

## Data Availability

The dataset is available from the corresponding author on reasonable request.

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
