# Peer review of "The Mother-Baby Bond: Role of Past and Current Relationships"

_children, 2023, doi:10.3390/children10030421_

Round 1

Reviewer 1 Report

Dear authors,

The topic is interesting, but you should more explicitely say about the novelty of your results and practical implications of them.

The following notes will also help to improve your manuscript quality and performance:

P.2, Lines 77-78: extra spaces to remove and missing spaces insert – line 89 (check these issues in the whole manuscript, you have both problems in several places).

P.2, line 83: Try to avoid using too many abbreviations, a reader might not keep everything in memory:

“Adult AS was assessed using the Relationship Questionnaire (RQ). Women were…”

P.4, line 160: “Tables” should start with a capital letter. And correct in other places also (P.6, etc...)

P. 4: Tables 1 & 2 and the rest of manuscript: Please, use the homogeneous style of separating decimals: “,” or “.”

P.6: Figure 1: In the title of the Figure is saying “plots” (plural), but you have only 1 plot.

Table 4: Both, parametric (r Pearson) and non-parametric (rho Spearman) are presented -> please, justify. In case if some variables are not distributed normally, the use of other parametric analyses is not coherent: t-student, ANOVA, etc..

Table 8 – ANOVA – the first line with titles is moved and not displaying correctly.

P. 13, line 303: “Considering again that our sample was not under mental health treatment,…”

The sample cannot be under treatment -> “participants of the study”…

General comments: Please, check the style of text in the Tables -> should be the same and as per the journal template indications.

Also try to represent the information concisely and choosing wisely either to present it in the Table on in the written text -> too many Tables (or unnecessary Figures). Less important (secondary information) should be placed in Annexes.

Author Response

Dear authors,

The topic is interesting, but you should more explicitely say about the novelty of your results and practical implications of them.

We thank the Reviewer. We substantially revised the discussion to make it clearer

The following notes will also help to improve your manuscript quality and performance:

P.2, Lines 77-78: extra spaces to remove and missing spaces insert – line 89 (check these issues in the whole manuscript, you have both problems in several places).

P.2, line 83: Try to avoid using too many abbreviations, a reader might not keep everything in memory:

“Adult AS was assessed using the Relationship Questionnaire (RQ). Women were…” 

We thank the Reviewer. We removed many abbreviations

P.4, line 160: “Tables” should start with a capital letter. And correct in other places also (P.6, etc...)

  1. 4: Tables 1 & 2 and the rest of manuscript: Please, use the homogeneous style of separating decimals: “,” or “.”

P.6: Figure 1: In the title of the Figure is saying “plots” (plural), but you have only 1 plot.

We thank the Reviewer. We corrected the typo

Table 4: Both, parametric (r Pearson) and non-parametric (rho Spearman) are presented -> please, justify. In case if some variables are not distributed normally, the use of other parametric analyses is not coherent: t-student, ANOVA, etc..

We thank the Reviewer. We have incorrectly entered parts of the exploratory analyses. The variables were normally distributed. Accordingly, we removed non-parametric analyses.

Table 8 – ANOVA – the first line with titles is moved and not displaying correctly.

We thank the Reviewer. We revised the Table

  1. 13, line 303: “Considering again that our sample was not under mental health treatment,…”

The sample cannot be under treatment -> “participants of the study”…

We thank the Reviewer. We revised the English of the entire manuscript

General comments: Please, check the style of text in the Tables -> should be the same and as per the journal template indications.

We double-checked the Tables

Also try to represent the information concisely and choosing wisely either to present it in the Table on in the written text -> too many Tables (or unnecessary Figures). Less important (secondary information) should be placed in Annexes.

We thank the Reviewer. We revised the Tables and the Results

Reviewer 2 Report

This paper aims to understand better the mother-child relationship that continues throughout pregnancy and postpartum. The study's methodology is well evaluated, with the survey period divided into four stages and the Edinburgh Postnatal Depression Scale and other instruments used to analyze the survey from multiple perspectives. In addition, the statistical methods used in each study are correctly described.

I recognize that this paper may be an essential one that suggests an approach to the period from pregnancy to postpartum, which is very important for constructing the mother-child relationship.

Major Comments

Structure of the Paper

In the organization of this paper, I believe that the following items should be reviewed, and the format of the research paper should be revised again. However, if the author intentionally designed the structure of this paper, please reject my recommendation.

In particular, I feel that the presentation of the tables and figures, even though they are essential data that have been thoroughly researched and accurately analyzed, is problematic. By correcting this, I believe the study's intent will be conveyed more clearly to the reader.

Missing Abstract

Uniformity of the format of tables in the paper (including notation of units, font, size, notation method, etc.)

Formatting figures in the paper (including notation of units, etc.)

Addition of Conclusion

I think the discussion's last paragraph should be modified to include a conclusion.

However, if you intentionally omit "conclusion," please reject my opinion.

In the following part, the author should clearly state which result is derived from this study.

" First, we reported that insecure attachment style of the mother was associated with the absence of exclusive breastfeeding. "

(L272-273)

Minor comments

In some paragraphs, there are several line breaks in the middle. Please check again.

(e.g., L113-114, L126-127)

Incorrect spacing between words and " (. "

(e.g., L115)

Spelling error? brestfeeding (breastfeeding ?) (L143)

Please double-check the notation in the text.

Author Response

This paper aims to understand better the mother-child relationship that continues throughout pregnancy and postpartum. The study's methodology is well evaluated, with the survey period divided into four stages and the Edinburgh Postnatal Depression Scale and other instruments used to analyze the survey from multiple perspectives. In addition, the statistical methods used in each study are correctly described.

I recognize that this paper may be an essential one that suggests an approach to the period from pregnancy to postpartum, which is very important for constructing the mother-child relationship.

We thank the Reviewer for the comment.   

Major Comments

Structure of the Paper

In the organization of this paper, I believe that the following items should be reviewed, and the format of the research paper should be revised again. However, if the author intentionally designed the structure of this paper, please reject my recommendation.

In particular, I feel that the presentation of the tables and figures, even though they are essential data that have been thoroughly researched and accurately analyzed, is problematic. By correcting this, I believe the study's intent will be conveyed more clearly to the reader.

Missing Abstract

    We are sorry for the typo, now we inserted the missing abstract and keywords

Uniformity of the format of tables in the paper (including notation of units, font, size, notation method, etc.)

Formatting figures in the paper (including notation of units, etc.)

   We thank the Reviewer for giving us the opportunity to improve the uniformity of the format of figures and Tables

Addition of Conclusion

I think the discussion's last paragraph should be modified to include a conclusion. 

However, if you intentionally omit "conclusion," please reject my opinion.

We thank the Reviewer. We have omitted the conclusion. However, we revised the discussion to clarify what our most significant findings and clinical implications were 

In the following part, the author should clearly state which result is derived from this study.

" First, we reported that insecure attachment style of the mother was associated with the absence of exclusive breastfeeding. "

(L272-273)

We corrected the sentence including the reference to the results. “First, we found that mothers with insecure attachment styles used bottle rather than breastfeeding at 1 month and 6 months postpartum (Table 5 and 7)”.

Minor comments

In some paragraphs, there are several line breaks in the middle. Please check again.

(e.g., L113-114, L126-127)

Incorrect spacing between words and " (. "

(e.g., L115)

We double-checked the spacing

Spelling error? brestfeeding (breastfeeding ?) (L143)

We corrected the word.

Please double-check the notation in the text.

We thank the Reviewer, we double-checked the notation in the text.